# Molecular and Genetics-Based Systems for Tracing the Evolution and Exploring the Mechanisms of Human Norovirus Infections

**DOI:** 10.3390/ijms24109093

**Published:** 2023-05-22

**Authors:** Sheng-Chieh Lin, Geng-Hao Bai, Pei-Chun Lin, Chung-Yung Chen, Yi-Hsiang Hsu, Yuan-Chang Lee, Shih-Yen Chen

**Affiliations:** 1Department of Pediatrics, School of Medicine, College of Medicine, Taipei Medical University, Taipei City 11031, Taiwan; 2Division of Allergy, Asthma, and Immunology, Department of Pediatrics, Shuang Ho Hospital, Taipei Medical University, New Taipei City 23561, Taiwan; 3Department of Internal Medicine, National Taiwan University Hospital, College of Medicine, National Taiwan University, Taipei City 10002, Taiwan; 4Division of Pediatric Gastroenterology, Department of Pediatrics, Shuang Ho Hospital, Taipei Medical University, New Taipei City 23561, Taiwan; 5Department of Bioscience Technology, Chung Yuan Christian University, Taoyuan City 32023, Taiwan; 6Center for Nanotechnology, Institute of Biomedical Technology, Chung Yuan Christian University, Taoyuan City 32023, Taiwan; 7Beth Israel Deaconess Medical Center, Harvard Medical School, Boston, MA 02215, USA; 8Broad Institute of MIT and Harvard, Cambridge, MA 02142, USA; 9Department of Infectious Diseases, School of Medicine, College of Medicine, Taipei Medical University, Taipei City 11031, Taiwan; 10Department of Infectious Diseases, Shuang Ho Hospital, Taipei Medical University, New Taipei City 23561, Taiwan; 11TMU Research Center for Digestive Medicine, Taipei Medical University, Taipei City 11031, Taiwan

**Keywords:** human noroviruses, genetics-based systems, evolution, recombination, reverse genetics

## Abstract

Human noroviruses (HuNoV) are major causes of acute gastroenteritis around the world. The high mutation rate and recombination potential of noroviruses are significant challenges in studying the genetic diversity and evolution pattern of novel strains. In this review, we describe recent advances in the development of technologies for not only the detection but also the analysis of complete genome sequences of noroviruses and the future prospects of detection methods for tracing the evolution and genetic diversity of human noroviruses. The mechanisms of HuNoV infection and the development of antiviral drugs have been hampered by failure to develop the infectious virus in a cell model. However, recent studies have demonstrated the potential of reverse genetics for the recovery and generation of infectious viral particles, suggesting the utility of this genetics-based system as an alternative for studying the mechanisms of viral infection, such as cell entry and replication.

## 1. Norovirus

Human noroviruses (HuNoVs) belong to the genus of norovirus in the family of *Caliciviridae* and are the predominant cause of epidemic and sporadic cases of acute gastroenteritis around the world [1,2]. Norovirus has a positive-strand RNA genome of approximately 7.5 kb, which contains three open reading frames (ORF) [3,4,5]. ORF1 encodes a polyprotein that can be cleaved into six non-structural proteins, including p48 [6], NTPase [7], p22 [5], VPg [8], Protease [9], and a viral RNA-dependent RNA polymerase (RdRp) [10]. ORF2 and ORF3 encode structural proteins—VP1 and VP2 [11]. Each viral particle capsid consists of 90 dimers of VP1 that includes a shell domain and a protruding domain, which serves as the viral ligand for binding to host histo-blood group antigens (HBGAs), acting as a membrane protein that acts as a receptor for norovirus [12,13,14]. VP2 is a minor structural protein which interacts with the interior surface of the capsid. VP1 is the major structural protein responsible for the interaction with host cells and evasion of the host immune response [11]. Noroviruses can be classified into six genogroups, which can be further divided into genotypes based on the genetic diversity of VP1 [15]. Noroviruses from genogroup I (GI), GII, and GIV account for human infections [16]. However, GII viruses are more commonly detected in patients with acute gastroenteritis [17]. The genotypes GII.3, GII.4, GII.6, and GII.7 of GII viruses are frequently detected in these patients. The genotype GII.4 suddenly emerged during the 2006/2007 season and caused a worldwide pandemic of gastroenteritis [18,19]. However, GII.3 was detected in infants and children with gastroenteritis, specifically in Japan, during 2003–2004 [20]. To date, the genotype GII.4 is predominantly responsible for the majority of clinical cases [21].

## 2. Evolution and Outbreaks

As described above, GII.4 is the norovirus strain of which we should be aware, since it contributes to major outbreaks in our societies [21]. Moreover, surveillance indicates the frequent emergence of new variants of GII.4, which replace previous strains and become predominant strains [22,23,24]. Significantly, increasing norovirus outbreak events are correlated to the emergence of novel GII.4 strains [25,26]. The molecular epidemiology of clinical cases shows that the frequency of new variant emergence is extremely high, taking place every 2–3 years [22]. These variants have been studied and shown to have a chronological relationship, which may be driven by selective pressures from herd immunity against older variants. The VP1 protein of these variants might combine with different polymerases, and the VP1 protein is responsible for binding to the host HBGA and immune evasion [11]. The alterations in VP1 of GII.4 lead to changes in HBGA binding specificity and an increasing preference for a specific host population, which may contribute to the high prevalence of specific GII.4 variants [27,28,29,30,31]. In addition to point mutation, recombination is also an important mechanism of norovirus evolution, which occurs between co-infected norovirus strains [32,33,34]. The major recombination site is positioned at the sequence between ORF1 and ORF2 that increases the relative genetic diversity [35]. In one study, the evolutionary rate of intervariant GII.4 strains was reported to range from 1.57 × 10^−3^ to 4.64 × 10^−3^ substitutions/site/year [30]. Among the variants, the Sydney 2012 variants had the higher substitution rate. GII.4 intervariant diversification, which might be driven by the immune status of the human population, is correlated with the accumulation of amino acid substitutions in the major viral capsid protein. Minimal randomly substituted amino acids located on the antigenic sites were observed with the predominance of each variant. The multiple pressures exerted on the virus, including virus–host interactions and dispersion, have been revealed to focus on the substitution of amino acids. Previous results suggest that the emergence of GII.4 variants may have three stages: the pre-pandemic stage, a short period of adaptability, and a pandemic phase [30]. This order of stages has appeared during the recent emergence and predominance of noroviruses in many Asian countries.

The current phylogenetic tree shows that a common ancestor of the GII.3 VP1 region diverged around 70 years ago and further diverged in 1984 and 1997. The evolutionary rates of the GII.3 VP1-coding region were estimated to be 4.82 × 10^−3^ substitutions/site/year [36]. Moreover, some positive and negative selection sites were found in the GII.3 capsid protein, functioning to drive the antigenic variations. In addition, the evolutionary rate of the VP1 region of GII.4 was higher than that of the GII.3 strain. The genetic divergence based on the phylogenetic distance of the GII.3 VP1 region was shorter than that of GII.4 [28]. These results suggest that the divergence of the GII.3 VP1-coding region may be relatively low.

The development of norovirus vaccines focuses on the major capsid protein, which is also the genomic region used for genotyping. Therefore, verifying the genotypes of GII.3, GII.4, and others for new variants is a priority, using powerful technology for detection. The vaccine candidates should protect against a broad diversity of genotypes and be easily adaptable to emerging genotypes. Due to noroviruses’ substantial contribution to the prevalence of diarrheal diseases among children and causing of more severe illness and death, childhood vaccination is more important for reducing norovirus prevalence among children. If vaccination prevents transmission, it might also reduce infections among all age groups. In summary, due to the high evolution rate of noroviruses, it is important to introduce emerging technologies so as to collect more comprehensive genetic diversity information of noroviruses from environmental samples and clinical specimens.

## 3. Technologies for the Detection and Analysis of Noroviruses

### 3.1. Nested Polymerase Chain Reaction (PCR) with Sanger Sequencing

The genotyping of norovirus has been performed via nested PCR, which uses genogroup-specific primer pairs to amplify of the VP1 region in complementary DNA (cDNA) samples to generate a template for Sanger sequencing (Table 1) [37]. Primer pairs for nested PCR have been specifically designed to amplify the VP1 sequence encoding the N-terminal domain and the shell domain of the capsid, and the resulting amplicon can be subjected to Sanger sequencing to identify the norovirus genotype in the samples [37]. Given its ability to provide sequence information, this method is widely used in clinical laboratories to monitor the evolution of norovirus.

### 3.2. Reverse-Transcription Quantitative Polymerase Chain Reaction Assays (RT-qPCR)

RT-qPCR assays are standard molecular diagnostics for detecting norovirus RNA in clinical specimens (Table 1) [38,39,40]. The advantages of RT-qPCR include its ability to detect norovirus RNA from various sample types, high sensitivity, and capacity for the identification of genogroups [40]. RT-qPCR can probe norovirus RNA from stool [40], water [41], food [42], and even environmental samples [43], which is important for evaluating potential transmission routes. Additionally, since norovirus is a highly contagious virus, the particle number for infection can be as low as 18–1000 copies [44]. RT-qPCR can detect as few as 10 copies of norovirus RNA and provide information on the viral load [45]. Furthermore, RT-qPCR uses different primer sets to detect specific genogroups [37], making it an efficient tool for identifying norovirus genogroups. However, compared to the nested PCR with Sanger sequencing method, RT-qPCR has a lower resolution and can only provide information on norovirus genogroups. In summary, RT-qPCR is an extraordinary technology for the rapid detection of norovirus and providing information on the viral load in samples. In addition, norovirus RT-qPCR assays have been incorporated into several emerging technology platforms, such as digital PCR and microfluidic multiplex PCR [46,47,48]. In particular, the microfluidic, multiplex PCR platform was developed for the detection of multiple gastrointestinal pathogens [47,48].

### 3.3. Digital Polymerase Chain Reaction (dPCR)

To improve the detection methods for low-level pathogen densities in samples, digital PCR is an alternative approach to quantitative detection from DNA. Digital PCR works by partitioning a unique sample into thousands of individual reactions running in parallel. Digital PCR can amplify target molecules that are calculated using Poisson statistics and do not need external reference standards. Two different digital platforms, the micro/nanofluidic-based and droplet-based approaches, are utilized for detection. The most widely studied platforms are the microfluidic-based Biomark^TM^ HD system (Fluidigm, South San Francisco, CA, USA) and the droplet-based QX100^TM^ and QX 200^TM^ Droplet Digital PCR (Bio-Rad, Hercules, CA, USA) (Table 1) [49]. Currently, it is impossible to perform a one-step digital RT-PCR (RT-dPCR) reaction using viral RNA. Compared to traditional quantitative PCR (qPCR), which has an insufficient sensitivity to quantify viruses, RT-dPCR allows for the quantification of norovirus GII and offers improved sensitivity compared to qPCR [49]. Several studies have reported the use of digital PCR to detect HuNoV RNA in samples from shellfish. Droplet dPCR (ddPCR) has greater precision in terms of quantification than RT-qPCR [50]. The application of ddPCR can provide accurate viral quantification for further risk analysis to ensure the safety of products on the market [51,52]. Triplex ddPCR can also perform simultaneous quantifications of norovirus GI and GII and hepatitis A virus in food, drinking water, and fecal samples, suggesting that ddPCR has greater sensitivity, accuracy, and anti-interference performance features than RT-qPCR [46]. Recently, a novel microfluidic-based ddPCR chip was developed for the absolute quantitative detection of HuNoV. The chip is based on digital PCR, and the sample solution is divided into microdroplets through microfluidic technology. The chip has a comparable sensitivity to ddPCR and may provide an alternative method for the detection of HuNoV due to its advantages of a high throughput and high sensitivity [53,54].

### 3.4. Enzyme Immunoassay (EIA)

The enzyme immunoassay (EIA) is the most common test for the rapid detection of pathogens in clinical practice. These EIAs can be applied in large-scale clinical and epidemiological studies. EIAs require antibodies to detect various norovirus genogroups and/or genotypes. Thus, EIAs are based on polyclonal or monoclonal antibodies used to monitor different virus-like particles (VLPs). Previous studies have demonstrated that the sensitivity and specificity of norovirus EIAs vary according to diagnostic goals (Table 1) [55]. Commercial norovirus EIAs are available for the examination of minute amounts of stool taken as samples. However, the sensitivity of EIA is lower than that of the other tests; therefore, EIA is only suitable for use as a companion test together with other tests, such as RT-qPCR, to increase the detection efficiency. A previous report showed that a positive signal of EIA was 4.20 × 10^8^ copies/g of fecal sample, which is equal to a cycle threshold value (CT value) of 25.6 based on the standard curve. Thus, fifty percent of GII samples might be false negatives based on EIA but show positive results with a CT value higher than 26.5 [55]. In addition, during the first 48 h of a norovirus outbreak, the viral load in fecal samples ranges from 10^7^ to 10^8^ copies/g [56]. This information suggests that samples collected later than 48 h after the onset of symptoms could yield negative results using the EIA method. However, additional studies are needed to evaluate the limits of detection of other genotypes. The advantage of the EIA method is that the kit has successfully detected 18 of the 21 norovirus genotypes evaluated. In summary, the EIA norovirus kit may be useful for the rapid screening of fecal samples collected during a norovirus outbreak of acute gastroenteritis, but the negative samples should be confirmed using a second technique, such as RT-PCR.

### 3.5. Next-Generation Sequencing (NGS)

Next-generation sequencing technologies offer a promising means to provide complete genome information for norovirus, which is important for investigating genetic diversity among strains, establishing evolutionary patterns, and tracing transmission chains in outbreak events. There are two NGS methods that can be used for analyzing the complete genome of norovirus: targeted sequencing [57,58] and metagenomic sequencing (Table 1) [59]. Targeted sequencing is based on the enrichment of sequences of interest through capture probes or primers to provide robust information on the genetic diversity of regions of interest [57,58]. The mutation frequency of norovirus is very high. Targeted sequencing can provide a precise mutation rate for specific regions in each sample, owing to the depth of targeted sequencing of specific regions [57,58]. In contrast to targeted sequencing, metagenomic sequencing provides genome information on all viruses in the sample rather than information on regions of interest from a specific virus. Metagenomic sequencing can help to study the relationships between co-infected norovirus strains, such as recombination or synergism [59].

A new strategy using a combination of NGS and third-generation sequencing (TGS) to provide highly accurate sequence information for isolated noroviruses was described [60]. The TGS platform Oxford Nanopore Technology can read up to 100 kilobases on a single DNA molecule and is an ideal method for evaluating recombination events and identifying of subgenomic sequences [60]. By combining the advantages of NGS and TGS, we can gain deeper insight into the genetic evolution of norovirus.

### 3.6. Aptamer-Based Detection of Specific Genotypes

Aptamers are mostly single-strand DNA or RNA oligonucleotides, such as antibodies, that can form structures to interact with specific molecules [61]. They are artificially synthesized using an in vitro technology known as the Systematic Evolution of Ligands by Exponential Enrichment (SELEX) (Table 1) [62]. Previous studies have developed several aptamer candidates that can specifically bind to GII.4, GII.3, and GII.7, respectively [63,64,65], and can also bind to different VLPs corresponding to various GI and GII HuNoV strains [66], suggesting that aptamers have significant potential for the development of genotype-specific assays and extraction methods that could be useful for the rapid identification of norovirus strains and enrichment of viral particles for further analysis [63,64,65,66]. However, like antibody-based methods, the sensitivity of aptamer detection or enrichment technologies might decrease over time because of the rapid mutation rate of noroviruses. In contrast to antibodies, the processes for the development of novel aptamers are faster and more affordable [67,68], suggesting that it is possible to cyclically select aptamers for the detection of emerging strains. Moreover, aptamer-based point-of-care detection is evolving, and lateral flow immunochromatographic assays and paper-based microfluidic devices have been developed for HuNoV detection [69,70]. The microfluidic device utilizes the fluorescence of the 6-FAM-labeled aptamer quenched by multi-walled carbon nanotubes (MWCNT) and graphene oxide (GO), which have potential for the rapid in situ visual determination of noroviruses [70].

### 3.7. Reverse Genetics Techniques

The HuNoV genome is a positive-sense, single-stranded RNA (+ssRNA) of ∼7.6 kb with three ORFs: ORF1, which encodes a nonstructural polyprotein, and ORF2 and ORF3, which encode the major and minor capsid proteins VP1 and VP2, respectively [71]. The absence of an in vitro framework hinders the study of the HuNoV life cycle, leading to a focus on utilizing other caliciviruses and murine norovirus (MNV) that can be cultured in mammalian cells [72]. A 3′ coterminal polyadenylated subgenomic RNA is created inside infected cells. Genomic and subgenomic RNAs have similar nucleotide arrangement patterns at their 5′ ends, which are covalently connected to the nonstructural protein VPg at the 5′ ends in HuNoVs, as has been demonstrated for MNV [73,74]. Nonstructural proteins are communicated from genomic RNA and form an RNA replication complex that creates new genomic RNA particles, as observed in subgenomic RNAs encoding VP1, VP2, and the interesting protein called VF1 during MNV infection of cells [8]. The capsid is gathered, and viral RNA is encapsidated before descendants’ discharge after the articulation of the underlying proteins from subgenomic RNA particles [8].

Reverse genetics for positive-strand RNA viruses infections depends on the gathering of full-length cDNA clones in a plasmid vector or, for larger viruses such as coronaviruses, in bacterial or yeast artificial chromosome vectors (BAC and YAC) and their engendering in microorganisms or yeast [75]. Fusion with the bacteriophage T7 or SP6 promoter enables the cell-free production of viral RNA with T7 or SP6 RNA polymerases. The fusion of a eukaryotic ubiquitous promoter, rather than T7 or SP6 promoters, induces the generation of viral RNA from transfected DNA via endogenous cell RNA polymerase II [76]. Reverse genetics techniques have generally been utilized in the RNA virology field, engendering full-length cDNA clones, especially for larger viruses with some popular arrangements among microscopic organisms (Table 1) [77,78]. A few methodologies have been developed to overcome these issues, including the utilization of extremely low-copy-number plasmids, change of enigmatic locales, production of full-length DNA formats for in vitro RNA recorded via the in vitro ligation of DNA fragments, and the cotransfection of a combination of covering DNA fragments with the principal section containing the eukaryotic articulation advertiser upstream of the viral 5′ untranslated region (5′ UTR) sequence [79,80]. Though useful for some positive-strand RNA viruses, these methodologies have either not been fruitful or not been attempted for most RNA viruses. They frequently require the development of customized conditions, such as the utilization of particular plasmid vectors and bacterial strains, the restricted selection of sections due to explicit areas of limitation destinations, and the need for large arrangement covers.

#### 3.7.1. Reverse-Genetics-Based Norovirus Inhibitor Screening (NoVIS)

For the development of a norovirus inhibitor screening (NoVIS) system, the HuNoV GII.4 genome (Figure 1a) can be cloned into a transcription unit with an elongation factor-1α (EF-1α) promoter, transcription binding sites for SP-1 and AP-1, and a ribozyme from hepatitis delta virus for the efficient transcription of the norovirus (NoV) genome with a 5′ cap and poly(A) sequence. To produce reporters for protease and RdRP in vivo activities, green-fluorescent protein (GFP) and Renilla Luciferase can be inserted into ORF1 and ORF2 (Figure 1b). For the quantification analysis of protease inhibition, the GFP CDS can be split into N-terminal and C-terminal fragments and inserted into positions in a protease digestion site within ORF1, which will generate a reporter system of protease-inhibitory-dependent fluorescence emission. Once protease is inhibited, it will not be digested, leading to the generation of a single polyprotein that contains intact GFP. The inhibition percentage can be correlated with fluorescence signals. Therefore, this system can be used to quantify the efficacy of potential protease inhibitors (Figure 2a).

For the detection of RdRP activity, since ORF2 expression is mainly dependent on the synthesis of subgenomic RNA, a Renilla luciferase CDS can be inserted into ORF2 to generate a fusion reporter protein that is expressed under the activity of RdRP. Hence, the activity of RdRP can be correlated with bioluminescence signals (Figure 2b). In conclusion, these fluorescence/bioluminescence reporters for norovirus protease and RdRP can be useful for studying the mechanism of the norovirus life cycle and for screening potential drugs for norovirus.

#### 3.7.2. Current Status of Reverse Genetics for Human Norovirus

A reverse genetics system can generate viruses from cloned constructs. Thus, it is a powerful tool that can be used to overcome the genetic diversity of norovirus and discover the pathogenesis of, and design antivirals for, norovirus [81]. Asanaka et al. first developed a mammalian-based replication system for a non-cultivable human norovirus. They used the vaccinia virus strain MVA to produce T7 RNA polymerase. The system then drove the expression of the norovirus viral RNA in the mammalian cells and packaged the viral RNA into virus particles [71]. Moreover, Katayama et al. established a plasmid-based reverse genetics system driven by a mammalian EF-1α promoter without a helper virus. The system can support HuNoV GII.3 viral genome replication and produce viral particles containing GFP-marked genomic RNA. The system is able to produce infectious progeny virions and is also generalizable to both human and animal noroviruses [82]. In addition, Oliveira et al. cloned HuNoV GII.4 Sydney subtype cDNA into a pcDNA3.1-based plasmid vector downstream of a cytomegaloviral promoter. They also inserted the cDNA into the replication subsystem of the GFP reporter gene. The replicon containing GFP was then transfected into human Caco-2 cells and expressed VP1/VP2 capsid proteins [83].

**Table 1 ijms-24-09093-t001:** Summary of technologies used for the detection of noroviruses.

Techniques	Detection Principles	Advantages	Disadvantages	Time	Cost	Application	Ref.
Nested PCR plus Sanger sequencing	Amplification of the VP1 region for Sanger sequencing	Highly sensitive, more high-quality and cost-effective than NGS	Time-consuming and labor-intensive	Moderately fast	Inexpensive	Provides sequence information and monitors evolution	[37]
RT-qPCR	PCR-based (nuclear acid amplification)	Highly sensitive; rapid results; compatible with various sample types	Lower resolution than sequencing; susceptible to contamination	Rapid	Inexpensive	Rapid detection and surveillance Known genogroup identification	[40]
RT-dPCR	PCR-based, microfluidic droplet platform	Improved sensitivity, precision, accuracy, and multiplexing compared to qPCR	More time-consuming and expensive than conventional PCR; technique expertise	Moderately fast	Inexpensive	Absolute quantification of norovirus	[49]
EIA	Antibody-based (VLP antibodies)	Rapid screening; simple and cost-effective Kits detect several different norovirus genotypes	Limited sensitivity and specificity Reliance on antibodies Escapes recognition from antigenic changes	Rapid	Inexpensive	Combined with RT-qPCR Negative samples should be confirmed a second time	[55]
NGS/TGS	Target and metagenomic sequencing	Highly sensitive and accurate High-throughput and comprehensive	Time-consuming, high-cost Requires high-quality samples	Lengthy	Expensive	Genetic diversity and evolution Recombination or synergism information	[57,58,59]
Aptamers	Synthetic oligonucleotides bind to target molecules (such as antibody-based molecules)	Faster and affordable More stable than antibodies Easy to modify and develop novel aptamers	Limited sensitivity and specificity Temperature affects binding capacity	Rapid	Inexpensive	High-throughput screening and diagnostics of emerging norovirus strains	[63,64,65]
Reverse genetics	Generates viral genomic RNA transcripts within a host cell	Eliminates interference from heterologous viruses Ensures the quality of RNA transcripts	Time-consuming and requires technical expertise Requires particular plasmid vectors and bacterial strains	Lengthy	Expensive	Viral genomic structure, virus–host interaction and pathogenesis, vaccine development	[75,76,77,78]

Abbreviations: RT-qPCR, reverse-transcription quantitative polymerase chain reaction assays; dPCR, digital polymerase chain reaction; EIA, enzyme immunoassay; NGS, next-generation sequencing; TGS, third-generation sequencing.

## 4. Recent Advances in Norovirus Evolution and Recombination

### 4.1. Norovirus Evolution

Human noroviruses are the leading cause of foodborne illness, causing both acute and chronic gastroenteritis. The human immune system is believed to be driven by the selection of emerging pandemic NoVs through both antigenic drift and shift. This phenomenon results in the replacement of dominant circulating viruses with new variants that are able to reinfect hosts who have previously been infected with earlier viruses [84]. The majority of NoV outbreaks worldwide are caused by the genogroup II.4 (GII.4), with new variants emerging every 2 to 4 years. Immunocompromised patients are hypothesized to be important reservoirs in whom new NoV variants emerge. A previous study suggested that novel variants of GII.4 with HBGA and antigenic site changes were produced in immunocompromised patients [85]. A recent study found that viruses in immunocompromised hosts are genetically distinct from viruses circulating in the general population. Therefore, these patients may represent a reservoir for newly emerging strains [86]. An investigation of the intra-host viral diversity presented by noroviruses during the acute and shedding phases of infection in children showed that the intra-host genetic variation during the shedding phase recapitulates the genetic diversity observed on the global level, particularly those mapping to the VP1 antigenic sites. This indicates the occurrence of intra-host evolution in healthy children and explains the source of norovirus mutations that result in diversification on the global scale [87].

The high prevalence of prolonged norovirus shedding and illness presents a model for in vivo molecular evolution in patients with severe immune dysfunction. It was found that only five amino acids mutated over 182 days, suggesting an immune-driven selection impact on patients and hospitals and the potential role of prolonged shedders as a reservoir for viral antigenic variants. This leads us to stress the importance of confinement outbreaks of NoV infection that occur in hospitals [88]. A time-scaled phylogenetic tree showed that the present GII strains diverged from GIV around 1630CE at a high evolutionary rate (around 10^−3^ substitutions/site/year), resulting in three lineages. The study showed that some linear and conformational B-cell epitopes were found in the deduced GII capsid protein. These results suggest that norovirus GII strains rapidly evolved with high divergence and adaptation to humans [89]. Based on the evolutionary trace (ET) algorithm for quasi-species dynamics and the study of the molecular evolution of human norovirus during chronic infection, residues within the P2 domain remained absolutely conserved in all partitions, presumably because there was no selective advantage to alter the HBGA receptor-binding specificity [90].

### 4.2. Recombination of Norovirus

Since 1996, NoV variants of a single genetic lineage, namely GII.4, have been associated with at least six pandemics of acute gastroenteritis and have caused between 62 and 80% of all NoV outbreaks. The emergence of these novel GII.4 variants has been attributed to rapid evolution and antigenic variation in response to herd immunity; however, the contribution of recombination as a mechanism facilitating emergence is increasingly evident. The occurrence of recombination in NoV is not uncommon. Homologous recombination events can increase the virulence of the virus and lead to genotyping mistakes in molecular epidemiological studies.

To identify recombination events, multiple sequence alignments were assembled from publicly available strains and were tested using the Recombination Analysis Tool (RAT), a recently developed software tool. Strains identified as putative recombinants using the RAT were further tested using a phylogenetic approach, LARD software (version 2001), and a Monte Carlo method to gain additional information on their status [91]. Recombination in the norovirus capsid gene may occur naturally, involving capsid domains presumably exposed to immunological pressure. Unreported recombination was detected in approximately 8% of norovirus strains, with recombination breakpoints mainly located at the interface of the putative P1-1 and P2 domains of the capsid protein and/or within the P2 domain. The recombination region displayed features such as a length, sequence composition (upstream and downstream GC- and AU-rich sequences, respectively), and predicted RNA secondary structure that are characteristic of homologous recombination activators [92]. A recent report showed intergenotype recombination among new norovirus GII.4 variants through the intergenotype recombination of NoV strains with a GII.4 capsid and a GII.P16 polymerase gene in the complete genome [93].

## 5. Outlook

In this review, we described the development of detection technologies for noroviruses. PCR-based methods, including nested PCR and RT-qPCR, are rapid, sensitive, and affordable, making them suitable for clinical laboratories. While these methods provide important information for the surveillance of noroviruses, PCR-based methods still have inherent limitations, such as their recombination and mutation rates. Antibody-based immunodetection methods are the preferred choice for the rapid detection of viral infections, which is essential for the immediate identification of pathogens during outbreaks. However, noroviruses have considerable mutation rates that might lead to antigenic changes in the P domains of major structural proteins, allowing them to escape recognition by antibodies. Next-generation sequencing is the major technology for analyzing noroviruses’ genetic diversity and investigating novel strains in outbreak events. The combination of NGS and TGS can provide more detailed information on recombination. However, the expense of NGS methods makes them unsuitable for regular detection. The aptamer, an affordable antibody alternative, has been shown to have high affinity toward specific genotypes. However, aptamer-associated technologies may be affected by the same issue as antibody-based methods. To surveil norovirus outbreaks, it is important to develop a point-of-care detection method which can not only rapidly detect the presence of noroviruses but also provide information on the genotype, which is important for preventing and controlling highly pathogenic novel strains.

## Figures and Tables

**Figure 1 ijms-24-09093-f001:**
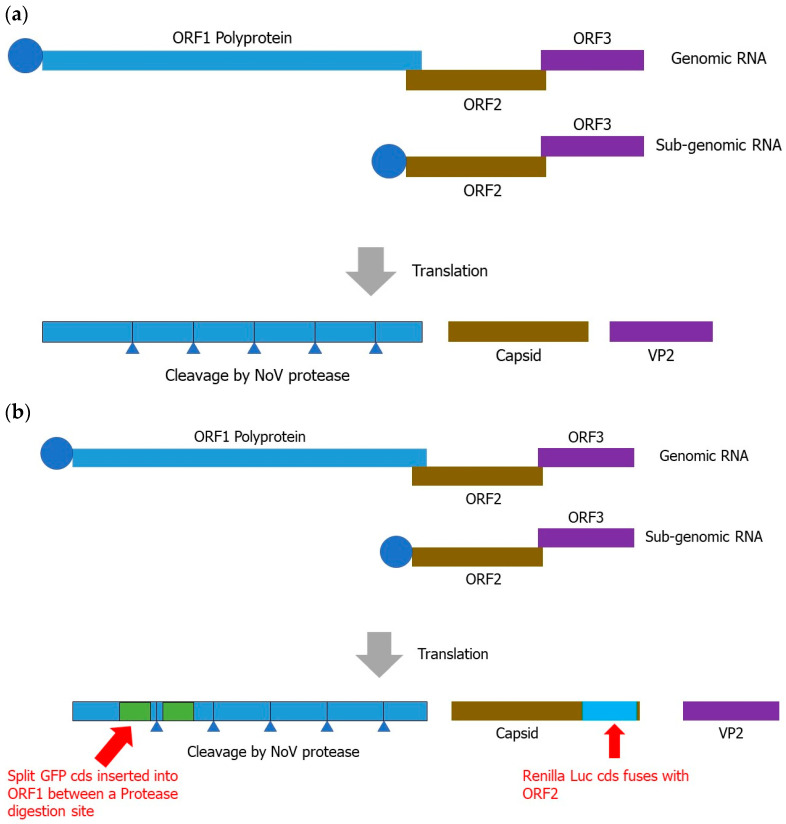
Scheme of norovirus and NoVIS system. (**a**) Norovirus genome and encoded polyproteins from ORFs. (**b**) Within the NoVIS system, GFP and Renilla Luciferase CDSs are inserted into ORF1 and ORF2, as demonstrated here.

**Figure 2 ijms-24-09093-f002:**
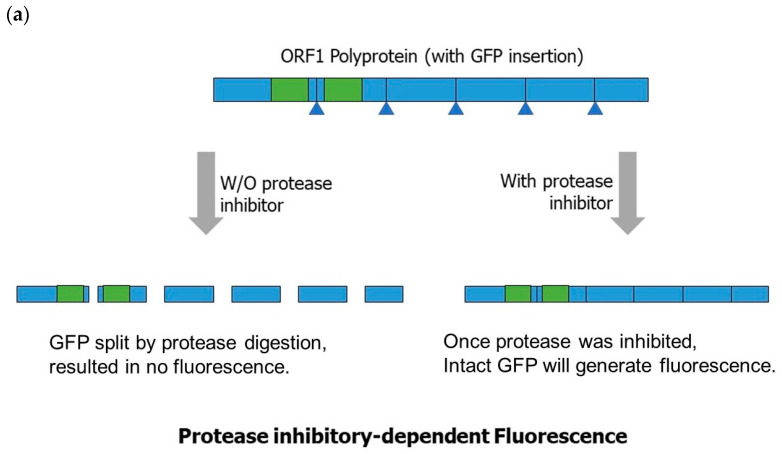
NoVIS system for the quantification of protease and RdRP activities. (**a**) Protease inhibition system used to quantify protease inhibitors. Blue regions indicate the polyprotein and digestion sites for protease. Two separated green regions are demonstrated as split green fluorescence protein. (**b**) Fusion reporter protein system for the detection of RdRP activity. Brown and blue parts are used to demonstrate the capsid ORF of Norovirus and inserted Luciferase. Red dot implies the relative light units from Luciferase, which decreases in the presence of RdRP inhibitors.

## Data Availability

Not applicable.

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
