# Peer review of "Molecular and Genetics-Based Systems for Tracing the Evolution and Exploring the Mechanisms of Human Norovirus Infections"

_ijms, 2023, doi:10.3390/ijms24109093_

Round 1
Reviewer 1 Report
This paper introduced Norovirus, its evolution, outbreak records, and technologies that can detect virus RNA mutation, virus infection and a serious of mechanisms related to this virus detection. Pretty detailed, but not that enough for one review. As these technologies are common used in Biochemistry, they are not specifically used for Norovirus and seems they are just randomly combined together for a paper.
For English wrighting it OK.
Author Response
Thank you for your comment. Our manuscript has been revised and restructured with earlier and updated content covering clinically applied methods for virus detection and evolution tracking. We have added several paragraphs, including the current status of the techniques mentioned for human norovirus and the development of point-of-care methods based on digital PCR and aptamers. We have also provided a table that presents the advantages, disadvantages, and applications of these methods for noroviruses.
Reviewer 2 Report
I’m very pleased to review this paper. Human noroviruses (HuNoV) are major causes of acute gastroenteritis around the world. This article reviewed the developments of detection technologies for noroviruses. PCR-based methods, including nested PCR and RT-qPCR, are rapid, sensitive, and affordable, which makes them being suitable for clinical laboratories. However, these methods providing particular important information for surveillance of noroviruses, PCR-based methods still have limitations inherent to them, such as recombination and mutation rates. So, it is important to develop a point-of-care detection method which can not only rapidly detect the presence of noroviruses, but also provide information of genotype, which is important for prevention and control of highly pathogenic novel strains.
I’m very pleased to review this paper. Human noroviruses (HuNoV) are major causes of acute gastroenteritis around the world. This article reviewed the developments of detection technologies for noroviruses. PCR-based methods, including nested PCR and RT-qPCR, are rapid, sensitive, and affordable, which makes them being suitable for clinical laboratories. However, these methods providing particular important information for surveillance of noroviruses, PCR-based methods still have limitations inherent to them, such as recombination and mutation rates. So, it is important to develop a point-of-care detection method which can not only rapidly detect the presence of noroviruses, but also provide information of genotype, which is important for prevention and control of highly pathogenic novel strains.
Author Response
Thank you for your comment and encouragement. Our manuscript has been revised and restructured with earlier and updated content covering clinically applied methods for virus detection and evolution tracking. We have added several paragraphs, including the current status of the techniques mentioned for human norovirus and the development of point-of-care methods based on digital PCR and aptamers. We have also provided a table that presents the advantages, disadvantages, and applications of these methods for noroviruses.
Reviewer 3 Report
The mansucript entitled "Molecular and genetics-based systems for tracing evolution and exploring the mechanisms of human norovirus infections" highlights importance of norovirus infections, its mechanisms, and its detection. Manuscript is well oragnised that requires minor editing of english.
NA
Author Response
Thank you for your comment. The English has been edited and the grammar has been corrected. The revised sentences have been marked in blue in the manuscript.
Reviewer 4 Report
Overly, it seems that this review is new and novel. The paper was well-designed and can be accepted. Anyway, the author can find some comments in the following.
It would be better to summarise the overview of all the techniques in one Table.
In Figures (1 and 2), the a and b sections were not presented.
Please improve the quality and graphic design of the figures.
Minor editing of the English language required
Author Response
Thank you for your comment. We have constructed a table to summarize the overview of all the techniques for the detection and analysis of noroviruses. The table is named Table 1 and can be found on pages 8-9 of the manuscript. In addition, the a and b sections are presented in Figures 1 and 2, respectively. Furthermore, we have improved the quality and design of the figures, and the new versions are on pages 6-8 of the manuscript.
Round 2
Reviewer 1 Report
After revision, this paper seems much better. It reviewed the norovirus and the methods of detectoin and tracking the virus infection. It becomes suitable to publish on the journal.
English writing is much better in the resion version.